# "[. . .] Un Tout Petit Peu de Dufayel"—Picasso, 1910–1914

## Laurence Madeline

Musées d'Arts et du Temps de Besançon, 25000 Besançon, France; laurence.madeline@orange.fr

**Abstract:** Picasso twice quoted the name of Dufayel, once in relation with the name of the Louvre and once for the same period of his career, between 1910 and 1914. This essay explores the universe created by the businessman Georges Dufayel in order to understand the role it played in Picasso's evolving cubism from that of analytic to synthetic.

**Keywords:** cubism; mass consumption; social positioning

## 1. Department Stores

In *Life with Picasso*, Françoise Gilot describes a long discussion with Picasso that might have occurred between the winter of 1943 and the spring of 1944: "On ne peut pas échapper à son époque. On prend parti d'un côté ou de l'autre; on est cependant dans la mêlée [. . .]. Nous [Braque et lui] avions même l'habitude de nous demander en riant: 'Peins-tu comme le Louvre ou comme Dufayel?' et l'autre répondait: 'Comme Dufayel, naturellement!'".[1]

In *Les Dames de Mougins*, Hélène Parmelin records another remark the artist made before 1964: "Quand on était avec Braque, on disait: 'Il y a le Louvre, et il y a Dufayel.' Et on jugeait tout avec ça. C'était notre façon de juger la peinture qu'on regardait. On disait: 'Ça, non, ça c'est encore le Louvre . . . Mais là, là, il y a un tout petit peu de Dufayel!'"[2]

Between 1910 and the summer of 1914, Georges Braque and Pablo Picasso were especially close, for as Picasso commented, "À ce moment-là, presque chaque soir j'allais voir Braque dans son atelier, ou ben il venait chez moi. Il fallait absolument que nous discutions du travail accompli pendant la journée. Une toile n'était finie que si chacun de nous en jugeait ainsi".[3]

## 2. About a Quotation

We all know what the Louvre is. The first French National Museum, opened in 1793, was established through the appropriation of former royal and aristocratic collections, succeeded by Napoleon's seizure and enriched, year after year, with gifts and acquisitions, finally becoming the largest museum in the world and the incarnation of the "universal" museum. As for Dufayel, to understate, this name is far less known. One vaguely knows that department stores once bore his name, and, thus, one can interpret Picasso's statements as a more or less ironic opposition between the "temple du Beau" and the "temple du Commerce" Salmon (2021, pp. 38–40). In this respect, the latter designates a site of poor taste, mass production, commercialism, and consumption, whereas the art made during the "cordée" with Braque (Bernadac 2021, pp. 70–71) could be a counter-reference to their own acts of artistic creation (Le Thomas 2007). In fact, other references to Parisian department stores appear in certain works by Picasso—for example, in *Le Rêve*, a drawing from 1908[4] representing two young women asleep in a pastoral setting, with the labels "Les magasins du Louvre" and "Au Louvre/Paris" glued upside down and on which is sketched *Le Pauvre Pêcheur* de Pierre Puvis de Chavannes;[5] "Le Bon Marché" and "La Samaritaine" appear in the pasted paper *Au Bon Marché* of 1913.[6] These raise questions about the significance of these references to the "cathédrales du commerce" (Zola 1883, p. 282) appearing within the field of the image.

And yet, these references to Dufayel, twenty years apart, that include Braque, do not particularly implicate other department stores such as the *Magasin du Louvre*, the *Bon Marché*, or the *Samaritaine*. Just as Marcel Duchamp bought his bottle holder in 1914 in the hardware section of the *Bazar de l'Hôtel-de-Ville*, and nowhere else—because he needed a store in the center of Paris and had an unrivaled reputation for DIY—Picasso summons Dufayel. This double reference to Dufayel requires consideration of his artistic approach at a crucial moment in his career as well as his social and economic circumstances: two discursive and material fields—those of artistic creation and those relating to social status or position within French society, which, like the Dufayel customers, are on the fringe of society but eventually reach actual social status.

The name of Dufayel must therefore be considered not as a particular individual retrieved from the memory of an artist of sixty or seventy years old, but as a character, a concept, perhaps a symbol of the period, who in the years 1910–1914, was proclaimed "l'homme du siècle".[7] In this respect, there is reason to consider whether Dufayel functioned for Picasso and Braque as kind of mirror or repoussoir in the evolution of cubism. Perhaps if we reflect on the trajectories of the businessman Georges Dufayel and Picasso's own fortunes, we may better understand the intersections of interests and objectives that are not simple oppositions (i.e., art versus commerce) but rather instances of the complex mechanisms and imbrications of capitalism, social relations, and commodification that are no less at stake in artistic creation.

### 3. About "Les Administatrions Dufayel"

Behind the name Dufayel lies the empire that this Parisian of working-class origins was able to establish in the last quarter of the 19th and beginning of the 20th century. Born in 1855, Dufayel began his working life in 1871 as a groom, or clerk, or sweeper, in a trading house belonging to Jules Crespin. Crespin, having first sold photographic portraits on credit to workers, opened his first shop in 1865 at 11–15 boulevard Barbès, in the Goutte d'Or district, (XVIIIth arrondissement). Subsequently, he offered, always on an installment plan, bicycles and sewing machines. Dufayel prospered in this firm and eventually became its sole owner in 1890. As a highly diversified enterprise, he named it the "Administrations et Grands Magasins Dufayel", which included the "Affichage national", the most powerful advertising agency in Paris, concessionaire of all advertising for the Eiffel Tower in 1889, followed by his firm's publicity for the Universal Exhibition of 1900. His company was also responsible for publications and displays for national elections, the distribution of newspapers and printed matter, and the publication of the Dufayel Indicator (a network of "Dufayel columns" mailboxes and simultaneously amid illuminated billboards). To these was added a comprehensive insurance company, offering life insurance and life annuities, as well the construction of the largest department store in the capital called the "Palais de la Nouveauté". At 25,000 m$^2$, it occupied four streets with more than 400 branches throughout France In 1897, when he was made a knight of the Legion of Honor, Dufayel, "comme un moderne empereur de Paris"[8], his company boasted a clientele of 1,790,600 customers and 19,400 employees (Figure 1).[9] His company also owned a seaside resort in Sainte-Adresse, the "Le Nice havrais", enhanced with a casino, marina, yacht club, three hundred villas, cottages, and apartments with access to the sea, available for purchase or rent. Additionally, he owned a stable of racing horses, participated in regattas with his yacht, and had a hotel built on the Champs-Élysées.[10] A paternalistic patron, he organized a yearly raffle every year that enabled one of his employees to win a furnished house and a garden in the suburbs, and he included his employees in his will, bequeathing them his stores.

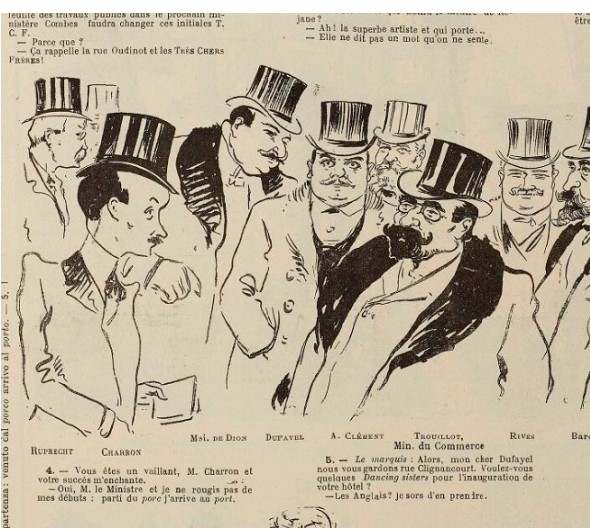

**Figure 1.** *La Chronique amusante*, 21 December 1905, p. 4. Paris, BNF, Département Philosophie.

### 4. Parvenu de Mauvais Goût

The annual raffle attests to Dufayel's awareness of the contemporary aspirations of the working classes, beyond the monthly credit often extended to them by local grocers, dairyman, and charcoal sellers. But with his savvy understanding of how the purchase of new commodities—bicycles, furniture, hunting rifles, trinkets, pianos, even vacations from urban labor—appealed to the working classes and by searching for them in every corner of the capital, he creates new consumers, a new class of micro-bourgeois who furnish themselves, settle in, and discover a certain comfort. Dufayel's progressive, albeit paternalistic beliefs helped create a new class of consumers—the micro-bourgeois[11]—who then gained access to previously unaffordable commodities. A survey of the budgets of Parisians revealed in 1909 that "[...] dans presque tous nos ménages parisiens, nous avons trouvé un compte Dufayel".[12] Jean Renoir, the filmmaker and son of the painter, knew one of these representatives of the Dufayel's enterprises, saying, "Le dernier des Paillepré était un homme charmant que mon père aimait beaucoup. Il était livreur chez Dufayel et parfaitement satisfait de son sort. Après son travail, il aimait fumer une pipe en regardant Renoir peindre dans le jardin. Il ne disait rien du tableau en train mais racontait volontiers sa journée, essayant de décrire les clients auxquels il avait livré des buffets Henri II ou une de ces lanternes imitation de fer forgé et à verres de couleur, chères à Courteline".[13]

Dufayel's paternalism was virulently denounced during a major strike which, in December 1905, brought a large part of his "Administrations" to a standstill. The strikers—excluded in 1916 from their patron's will—revealed the inquisitive or acquisitive practices of certain managers and the low wages of Dufayel's salaried workers. A few years after this strike, a "dismissed employee", M. A. Bodéchon, published an open letter whose title, "Dufayel. Tourmenteur des pauvres. Parvenu de mauvais goût" (Bodéchon 1908), sums up the resentment provoked by Dufayel's Champs-Élysées mansion, his yachts, his prodigality, his contempt for his exploited employees, and those of his eternal debtors.

The condemnation of Dufayel by Bodéchon demonstrates the opprobrium, followed by the oblivion, into which his empire eventually sank. This in turn explains why Picasso's published statements seem so enigmatic, especially since French fiction has appropriated Dufayel as the symbol of bad taste, conspicuous consumption, or as a symbol of social decline exemplified by the kind of furniture and other goods he sold.[14] Stories featuring young women who rise to the status of courtesans are often characterized by their moving into one of Dufayel's furnished apartments. Thus, the boulevard play *L'École des Cocottes* by Paul Armont and Marcel Gerbidon recounts, "[...]l'ascension d'une petite dame de Montmartre, qui passe de la chambre meublée à l'appartement de chez Dufayel et à l'hôtel particulier rempli de bibelots sélectionnés" (Derys 1919).

Similarly, it is worth mentioning the way in which Jean Lafranchis, biographer of Louis Marcoussis, describes Eva Gouel, Picasso's future companion, who bears the pseudonym Marcelle Humbert: "En 1907 [Marcoussis] fait la connaissance à Montmartre de Marcelle Humbert. Marcelle a des aspirations bourgeoises [. . .]. Sa passion pour Marcelle l'emporte sur son amour de la peinture, il cède . . . Ils peuvent bientôt quitter la rue Lamarck pour un appartement plus grand, plus bourgeois, 33 rue Delambre, et, consécration de cet embourgeoisement, commandent un mobilier Dufayel complet".[15] Counterbalancing this disdain of the Dufayel system, there are other, more edifying stories, such as that of Florise, this "[. . .] young worker who complained of being a victim of [sewing] entrepreneurs" and who, thanks to the director of Printemps, Mr. Jaluzot, secured a job and "[. . .] se haussa d'un cran dans l'échelle sociale. M. Dufayel facilita l'acquisition d'un mobilier et d'une machine à coudre. Et peu à peu Florise se modifia, évolua . . ."[16] And, in 1948, while the "Palais de la nouveauté" was being systematically dismantled, Picasso returned to the memory of Dufayel.

We can locate a humorous drawing from 1900, depicting two men chatting in front of the *Goutte d'Or* department store: "Parait que le grand Turc se sert chez Dufayel! Parbleu! Il doit acheter des femmes . . . à temperament"—and raise that the self-made man left behind him "[. . .] une foule d'œuvres de bienfaisance, de sociétés de secours mutuels [qui] reçoivent des legs impornts. Jusque dans la mort, Dufayel a pensé aux humbles, aux modestes, à leur avenir difficile."[17]

The example of Eva (probably invented) and that of Florise (more credibly) illustrates the way in which Dufayel could be either celebrated for his paternalism—if not philanthropy—but praised for his production of affordable commodities for the working classes.

## 5. Democratization?

This characterization of Dufayel as an "upstart" and "nouveau riche" is simultaneously reinforced and inflected by other aspects of his character and entrepreneurial policies. Dufayel's tastes were generally conventional. Although he collected paintings by Largillière, Vigée-Le Brun, Van Loo,[18] and Corot[19], in other respects, his tastes were not that distant from those of his business friend, Ernest Cognacq, the founder of "La Samaritaine", whose hotel particulier on the Champs-Élysées was decorated floor to ceiling by fashionable artists (Henri Gervex in particular).[20] The "Dufayel style" tended towards historicism, exemplified by the "Henri II" dining room and the "Aubusson" or "Beauvais" salons (in imitation of Louis XV style). Dufayel's style also demonstrates, through its commercial offerings, a general desire to make comfort affordable and beyond access to culture, according to a paradoxical commercial offer. Dufayel offered his clients "pleasant attractions: cinema—from 1896—and concert, palmarium, reading room, five o'clock tea"[21]; he provided them with leisure and commodities, the quality of which was widely praised. The buffet was provided by the famous caterer Potel and Chabot, and the cinema was judged "Le plus parfait des cinématographes", whose productions "[. . .] are performed for the most part by the best artists from the *Comédie-Française*, the *Odéon* and the main theaters of Paris . . ."[22] "Le cinéma gratuit était une autre des audacieuses innovations de Dufayel", Jean Renoir, whose first experience of cinema occurred there, recalled years later.[23] And, noting the success of cinematography, the chronicle notes, "Tout le monde sait que le bon ton, décidément abandonné par notre vieille noblesse française, s'est réfugié dans l'établissement de [Dufayel], qui a su faire de ses magasins les derniers salons où l'on cause".[24]

The democratization of the photographic portrait, already established by Crespin, was extended by the mysterious invention of the "Lino-Peinture Dufayel" by Francis Pierre Petit (which is not referenced in the French copyright database),[25] putting the decorative within everyone's reach and challenging the artisanal by employing mechanical processes. At the "Palais de la nouveauté", Dufayel also endeavored to exhibit the styles of national palaces, with a series of models created by the painter Marcel Jambon, and to create, thanks

to a competition, a "style that synthesizes the times in which we live".[26] Even on occasion, painters like Maurice Utrillo "[…] copient des bandes chez Dufayel …",[27] perhaps via decorative strips.

This is the Dufayel paradox. It opens access to culture to the working classes but promotes a formatted, normative, academic taste, excluding this modernity of mass consumption (objects, posters, advertising) from which it makes its fortune and by which already challenges poets and artists.

## 6. Incompatibilities

Nothing attests to Picasso having walked through the naves of the Dufayel department stores, gradually built between 1892 and 1912. As Louis Boyer claims in his lament "De place en place", created in 1905, there existed a fundamental incompatibility between the life of an artist and the Administrations Dufayel:

"La peinture c'est beau mais c'est triste,

Ça manque un peu d'essentiel,

Faut pas compter sur un artiste

Pour se meubler chez Dufayel …"[28]

Moreover, Fernande Olivier, Picasso's companion during the beginnings of Cubism, describes their purchases on credit from small merchants in La Butte, the rare outings to the Saint-Pierre market to acquire a red shirt with white polka dots, and purchases from second-hand dealers in Montmartre at the time of the establishment on Boulevard de Clichy in 1909 (Olivier [1933] 2001, pp. 41, 43, 174). Compared with the efflorescence of the new department stores, this was an almost anachronistic mode of consumption. Nevertheless, when temporarily separated from Picasso at the end of the summer of 1907 and installed on rue Girardon in Montmartre, Olivier confided to her guests, Alice Princet, Germaine Pichot, Alice Toklas, and Gertrude Stein, "[…] les inconvénients qu'il y avait à acheter un crédit un piano et un lit."[29] Did Fernande take out an account with Dufayel? Writing her memoirs at the very beginning of the 1930s and being a guarantor of the mythology of earnest bohemianism while resisting the seductions of capitalism, Fernande was able to erase the controversial Dufayel from her memories.

The artists Zyg Brunner, Jules Depaquit, and Gino Severini (2011) clearly took out a subscription and were never able to repay their debts.[30]

Braque, moreover, who lived near the "Palais de la Nouveauté" and was fascinated by the exploits of Wilbur Wright, likely attended the Dufayel cinema to view the film dedicated to the exploits of the aviator screened in the fall of 1908, as he was able to observe his biplane at the Air Show on 3 December 1908, and was able to laugh at the review, *Gratte-Lune*, given at the "Little-Palace", evoking the famous adventurer of the air.[31]

## 7. Modernity

Whatever the case, Picasso was, like Braque, daily confronted with the name Dufayel displayed on most of the walls of Paris, with advertisements produced by the "Affichage national" (Figures 2 and 3). An omnipresent and aggressive display, which prompted a lawsuit against his firm in 1896 because of the colors—blue and red—of one of his buildings, was blamed for altering the goods of the shops around. The debates that arose then took note of the existence of a new urbanism: "[…] Paris, le Paris industriel, le Paris des affaires […] renferme bon nombre de maisons peintes: les déménageurs sont en jaune et les marchands de couleurs émaillent des nuances les plus disparates de gigantesques enseignes."[32] New urban planning within the limits of the capital, especially the opening of the tram lines, enabled Dufayel to extend his advertising network, which ultimately contributed to the shaping of the contemporary landscape. However, Dufayel recognized neither his "creation" nor "[…] l'obsédante beauté des inscriptions commerciales, des affiches […]"[33] and his historicizing styles of bygone eras. By tracing the contours of Dufayel's empire in this way, we can better chart the movements of a modernizing cul-

ture, which oscillated between the reassuring traditionalism of historicist furniture or architecture and a provocative modernity. One can also observe the shifts in working class and bourgeois culture that were manifested in the vigorous surges in consumerism transforming French society.

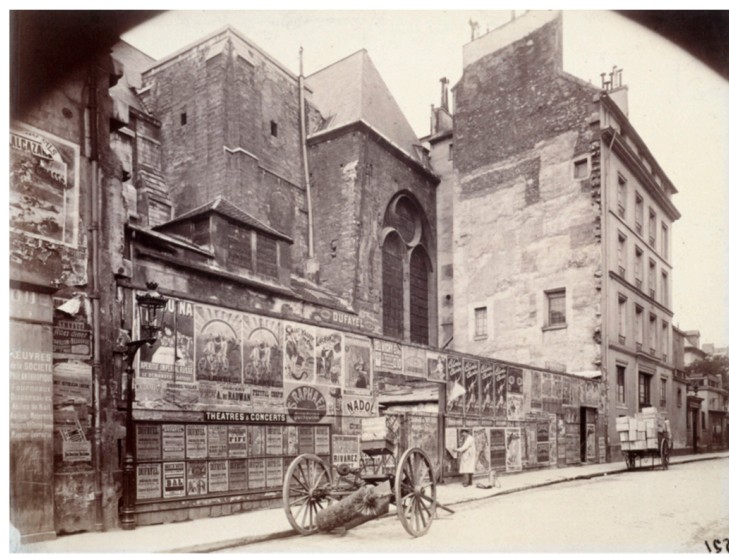

**Figure 2.** Eugène Atget, *Rue de l'Abbaye, Saint-Germain-des-Prés*, 1898. Paris, BNF, département des Estampes et de la Photographie, Estampes.

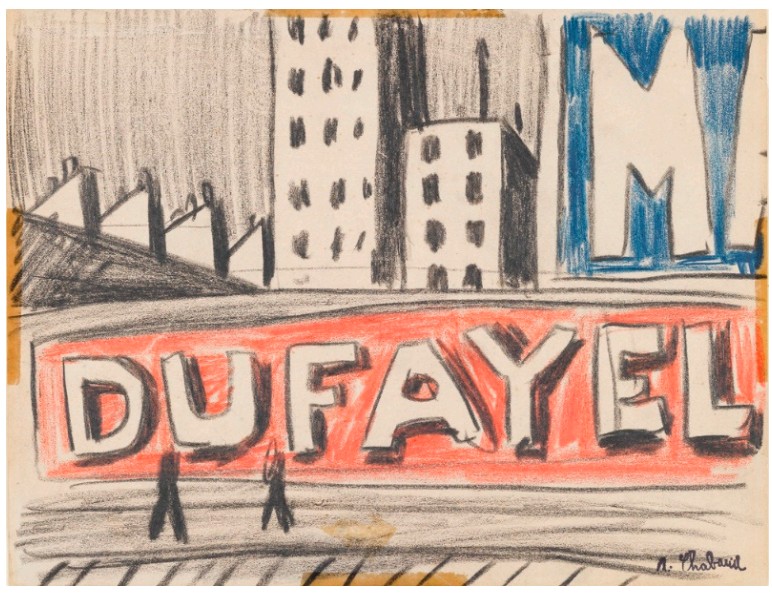

**Figure 3.** Auguste Chabaud. *Dufayel*, 1907–1912, Marseille, musée Cantini.

The ambivalence of the Dufayel business underlies Picasso's reference two or four decades later, of this already forgotten name, as well as the ambiguity of his declarations. There is laughter (contempt?) in the memory reported by Gilot. But there does not seem to be any on the part of Parmelin. Perhaps because ambivalence (being a bohemian or a petit bourgeois; being classical or introducing modern vocabulary in his work) is at the very heart of the work of Picasso and Braque, and Dufayel is perhaps a symptom or symbol of this ambivalence.

## 8. Standardization

When the two comrades refer to their relationship with Dufayel's enterprises (which have little to do with painting as such), this might parallel Picasso's own process of social advancement from precariousness towards stabilization—from the milieux of the lower classes to that of the bourgeoisie—on which the businessman based his success.

In September 1909, the artist left the Bateau-Lavoir to settle in a large comfortable studio in an artists' city on the other side of the Boulévard de Clichy, in the 9th arrondissement, a quarter that housed previous generations of artists and well-established theaters and whose population was primarily bourgeois. Fernande Olivier remarks on "Picasso s'embourgeoise", and also notes that "Il fallut acheter des meubles . . .", "Le style Louis XIV semblait avoir sa préférence . . ." (Olivier [1933] 2001)

The corresponding furniture style seems to have been standardized under Dufayel's tenure and must also be raised in the context of the 1910–1914 years.

In the photograph, he shot of a corner of his studio, *Composition photographique "Nature morte sur un guéridon"* (Figure 4); Picasso juxtaposes various objects on a fringed placemat. In the *Mandoliniste*, a hardly readable canvas from the fall of 1911, in *La table de l'architecte*, from the spring of 1912,[34] or in other paintings from the same period, Picasso depicts an extensive range of decorative and upholstery elements: tassels, braids, fringes, tiebacks, and ropes—so many motifs that, in their excess, parallel the series of *Intérieurs parisiens* photographed by Eugène Atget in 1910. The "Intérieur d'un employé aux magasins du Louvre " (Figure 5), with its pompoms and its studs (and its Henri II style furniture, probably purchased from Dufayel); the "Intérieur d'un ouvrier: rue Romainville" (Figure 6), with its wallpaper and fringes; and the "Intérieur d'une ouvrière rue de Belleville", with its accumulation of heterogeneous objects (Atget 1910) are exemplary of the taste and style of the more prosperous population of the petite bourgeois. Picasso proceeds with a search for overabundance, which is that of the period and particularly of the taste of these new emerging classes, which Atget captures in his *Interieurs* (4, 5, 6).

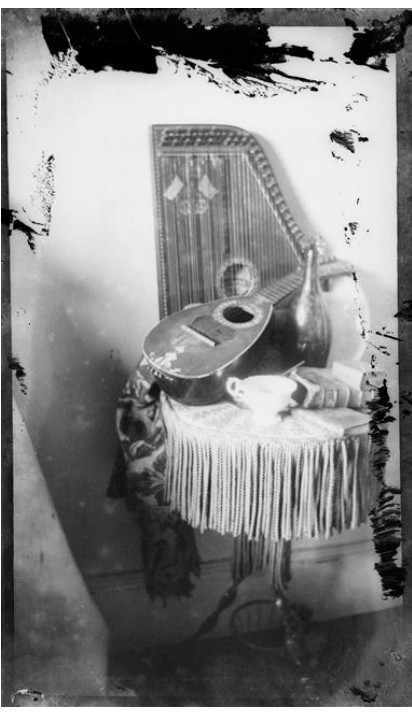

**Figure 4.** Pablo Picasso. *Composition photographique 'Nature morte sur un guéridon'*. 1911. Paris, musée national Picasso.

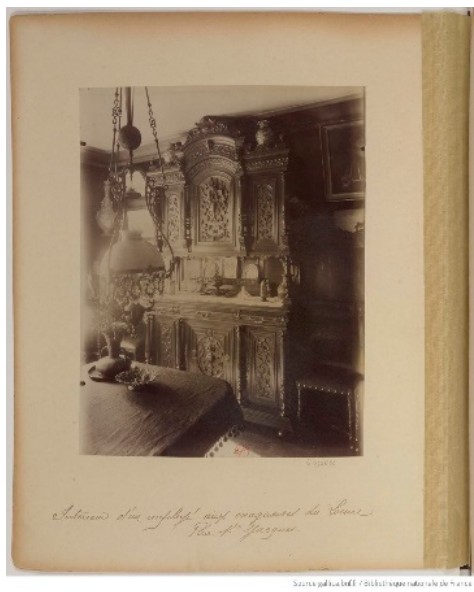

**Figure 5.** Eugène Atget, *Intérieurs Parisiens, "Intérieur d'un employé aux magasins du Louvre"*, Paris, 1910–1911. Paris, BNF, département des Estampes et de la Photographie.

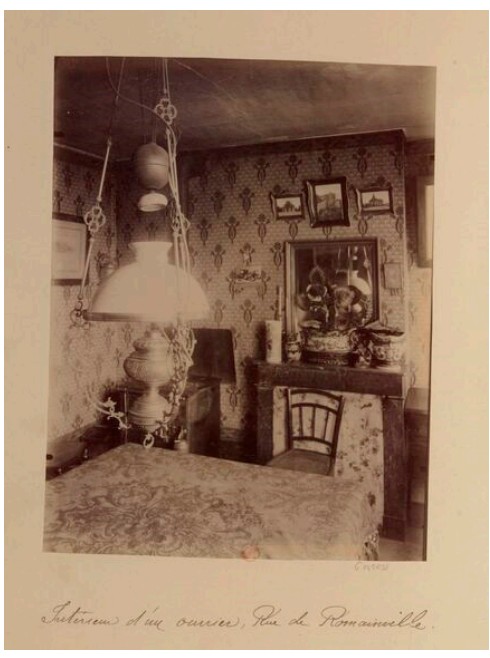

**Figure 6.** Eugène Atget, "*Intérieurs Parisiens, Intérieur d'un ouvrier: rue Romainville*", Paris, 1910–1911. Paris, BNF, département des Estampes et de la Photographie.

The essential Henri II buffet, popularized by Dufayel, was found in 1936, according to Marie-Thérèse Walter who remembers that on their rental villa in Juan-les-Pins, Picasso "[...] especially painted the Henry II sideboard that we had".[35]

### 9. Objects

Beyond remarking on the eclecticism and the glut of objects that Picasso observed in Dufayel's marketing, he further observed that "[...] les objets qui entrent dans ma peinture ne sont pas du tout comme cela: ils sont usuels: une cruche, un verre de bière, une pipe un paquet de tabac. Je ne vais pas chercher un objet rare dont personne n'a jamais entendu parler ..."[36] Dufayel's advertisements were ubiquitous in the daily press,

and featured "meubles, vêtements, phonographes, pianos, batterie de cuisine et œuvres d'art";[37] "[...] immense choix d'articles pour cadeaux en bijouterie, orfèvrerie, horlogerie, joaillerie, bronzes, objets d'art, petits meubles, maroquinerie, armes de chasse ..."[38]—a bric-a-brac where the everyday object and the exceptional are unified in their shared status as commodities. Here, Picasso opposes Dufayel's extravagant excess by reducing the objects featured in his still-life paintings. If, in the pile of quasi-indeterminate objects in *Bouteille, verre et fourchette or de L'Étagère*,[39] for example, we can perceive the timid evocation of the way stores displayed their goods, as in current unpacking, Picasso remains faithful to an asceticism on the margins of the commercial practices of his time.

## 10. Mass Production

It is still important to consider, from the spring of 1912, Picasso's use of sheets of wallpaper, linoleum and "[...] le faux bois, le faux marbre, le papier collé, tous ces 'éléments tout faits'..."[40] as a synecdoche of mass production, of the simulated as opposed to the authentic or the artisanal.

*Violon: Jolie Eva*[41] thus depicts a surface almost saturated with wood, where a garland and a wooden frieze stand out as those produced in commercial furniture factories. These were factories whose centuries-old organization was threatened by the Dufayel system. To meet massive demand, Dufayel's manufacturing practices accelerated the increase in production by lowering the cost of furniture, leading to drastic changes in the qualities of wood, which was increasingly thinner, employed more and more plywood, utilized prefabricated ornaments (garlands, friezes), and promoted simulated versions of traditional materials and illusionist processes (du Maroussem 1892).

As the mass production of wooden furniture multiplied (for dining rooms, bedrooms, living rooms), Picasso utilized illusionistic imitations of wood veneers in his still-life work. The use of machines and the specialization and division of labor that accompanied the increase in the productivity of workshops challenged the status of the artisan as a creator. This might be analogized to Picasso and Braque's attempts to renounce individualism by no longer signing their works (Gilot and Lake [1964] 2006). By using (or duplicating) manufacturing processes that supplanted artisanal techniques, they reflected on their own creative process, evoking the "*l'impersonnel*" (Kahnweiler 1963), identifying with those traditional artisans who would eventually be replaced by the waged labor and assembly lines of industrialized factories and the standardization of production.

## 11. Advertising

The Dufayel system made visible in advertising signs was ignored by Picasso in his work until the summer of 1912, when in Sorgues, he painted the *Paysage aux affiches*,[42] demonstrating a juxtaposition of factories treated in a quasi-hermetic cubism style, ocher and white, and including three aggressively colored patterns, as well as a bottle of Pernod, borrowed from a usual bistro table, an advertisement for Léon torn from a newspaper, and a KUB advertising billboard. We are far from the visual abundance mocked by the humorous magazine, *Le Rire*, showing on the front page of the issue of 16 August 1902, "Un peu de publicité", a garish superposition of posters stuck in a street corner and dominated by the brand "Dufayel" (Figure 7).

Picasso still remains in the background. He does not completely identify with the aesthetic soon to be advocated by Guillaume Apollinaire as "[...] des lettres d'enseignes et d'autres inscriptions parce que, dans une ville moderne, l'inscription, l'enseigne, la publicité jouent un rôle artistique très important ..."[43]

Dufayel's publicity is acknowledged but only moderately, as if the artist was always looking for moderate, classical construction and wanted to remain, above all, an artist organizing perennial artistic forms away from the visual crush.

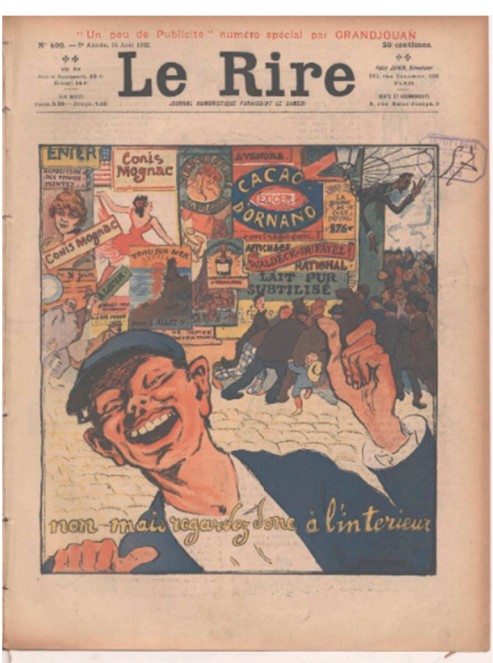

**Figure 7.** Grandjouan, "Non mais regardez donc à l'intérieur!", *Le Rire*, 16 août 1902, p. 1. Paris, BNF.

## 12. Picasso versus Dufayel

Thus can be explained the paradoxical positions of the artist and the businessman.

Dufayel shakes up social organization and participates in Parisian modernity (the "king-object"; the poster), but at the same time, he attempts to promote a reassuring traditionalism with his Henry II buffets, his actors from the national theaters, his orchestra, his academic painters, his Corot . . ..

Picasso who, while pulverizing the codes of pictorial representation, maintains a popular tradition of the still life, with its traditional accoutrements—the mandolin, the absinthe bottle, the beer glass—versus the bicycle, the sewing machine, and the visual shocks of the capitalist industrialization incarnated by Dufayel, clings to an already outdated way of life as described by Fernande Olivier.

Dufayel sought to bring to the working classes his notions of classic French taste—accessible, understandable, and reassuring. He thus combined the pleasure of aesthetic contemplation with that of money; "Art is business!" was offered as a motto in 1908.[44]

Picasso exhibited his works only through Daniel-Henry Kahnweiler's confidential gallery and sold, at quite high prices, his work to a very few amateurs; his hermetic paintings more and more seemed to be inspired by the world recomposed by Dufayel.

Dufayel appropriates and rationalizes the classic taste shaped by generations of aristocrats or bourgeois in order to sell it to the working classes, whom Picasso snubs.

Dufayel reproduces and duplicates landscapes using new techniques (photography, Lino-Peinture) and democratizes art.

Picasso would divert the technical processes of mass production to democratize artistic activity (Le Thomas 2016, p. 365).

Picasso, above all, transforms and transcends popular taste to remake or transform it for the enlightened haute bourgeoisie.

Perhaps, at this time, Picasso (and Braque) needed more of a Dufayel approach to art rather than the one provided by the Louvre, because the world shaped by the businessman was more modern, and the Louvre needed to be challenged by another aesthetic entity that was, eventually, more productive and more social. Additionally, Picasso himself needed to step out of Montmartre and the traditional way of living in order to confront the current modern world.

### 13. Jean-Baptiste Camille Corot

An intersection between the trajectories of the two Cubist painters and Dufayel might be located (however improbably) around a painting by Jean-Baptiste-Camille Corot. In the spring of 1910, Picasso and Braque each executed a *Femme à la mandoline*.[45] Judy Cousins compares them to Corot's *Femme portant une* toque,[46] exhibited at the Salon d'Automne of 1909, which by then had been elevated to a to a recognized masterpiece of French painting, the inheritor of a classicism which was considered both sensual and natural. Reproduced in the *Gazette des Beaux-Arts* of 1 July 1909, it was, in fact, owned by Dufayel. Corot thus appears as a shared and admired ancestor by both old magnate and young Cubists, as a respected representative of French, classical painting, and successor of Jean-Auguste-Dominique Ingres, who Picasso admired as well. In 1910, thanks to a trade with Wilhelm Uhde, the painter acquired a painting by Corot, which joined the "primitive objects"—Iberian, African, Oceanian—and the Douanier Rousseau works in his own collection.[47]

### 14. Something in Common?

After leaving the Butte Montmartre, located at "[…] la lisière de la grande ville, entre campagne cultivée et friches, […] lieu favorable à la persistance de traditions paysannes et rurales anciennes …"[48] and embracing a less bohemian existence, Picasso seems to have opened his work to the social and cultural disruptions of the city embodied by Dufayel and from which he had previously kept his distance. The vulgarity associated with Dufayel's productions—as when Picasso indicates his disdain in his remark to Parmelin: "Il faut savoir être vulgaire"[49]—are a denunciation of Dufayel's uninventive taste, his deployment of his commercial genius to influence the "great mass of the people" in the service of consumption and commodity fetishism, and his influence in redesigning the urban landscape. Dufayel becomes one of the architects, therefore, of the openly consumerist modern society. It is ultimately what allows Picasso to be part of his time, to place himself—even fleetingly, and after a long attachment to the social organization of the end of the 19th century—on the side of the artists taking responsibility for the surrender in question of social organization and traditional aesthetics through consumption. "In some important respects, the Cubists seem to have made choices that were willfully regressive, or at least against the grain of what many thought was progressive", and this is rightly reminiscent of Kirk Varnedoe and Adam Gopnik in *High and Low* (Varnedoe 1991).

To place "un peu de Dufayel" in his works from this period of his life, where his intense relationship with Braque was woven, as well as his moves and the fascinating inflection of Cubism—from an analytical Cubism towards a synthetic Cubism—the explosion of the frame and painting, etc., is to also recognize everything that the Louvre still weighs in its creation and the perennial duality that Picasso cultivates between proletariat and bourgeoisie, between scholar and popular, between elitism and mass.

A duality which is essentially that of Dufayel.

**Funding:** This research received no external funding.

**Data Availability Statement:** Not applicable.

**Conflicts of Interest:** The author declares no conflicts of interest.

### Notes

[1]　"'Do you paint like the Louvre or like Dufayel?' and the other replied: 'Like Dufayel, of course!'"(Gilot and Lake [1964] 2006).

[2]　"When I was with Braque, we used to say: 'There is the Louvre, and there is Dufayel.' And we judged everything like that. It was our way of judging the painting we were looking at. We said: 'That, no, that's still the Louvre … But there, there, there is a little bit of Dufayel'". Parmelin (1964, pp. 138–39).

[3]　"At that time, almost every evening I went to see Braque in his studio, or he came to my place. We absolutely had to discuss the work accomplished during the day. A painting was only finished if each of us judged it so." Gilot and Lake (Gilot and Lake [1964] 2006, p. 74).

[4]　Pablo Picasso, *Le Rêve*, Paris ou La Rue au Bois, 1908, plume, encre, gouache et étiquette sur carton, Berne, collection particulière.

5    Pierre Puvis de Chavannes, *Le Pauvre pêcheur*, 1881, Paris, musée d'Orsay. Acheté par l'État in 1887; the painting was still then in the collections of the musée du Luxembourg.

6    Pablo Picasso, *Au Bon Marché*, Paris, hiver 1912–1913, huile, papiers collés sur carton, Cologne, museum Ludwig.

7    *La Vie parisienne*, 13 January 1917, p. 42.

8    *L'Intransigeant*, 1 March 1907, p. 4.

9    See the chart of the Légion d'honneur n° LH/831/24, base Léonore, Archives Nationales.

10    Built in 1905, the hotel was destroyed in 1924.

11    On the opening of mass consumption to workers and the role of Dufayel, see (Wemp 2010).

12    "[. . .]. in almost all of our Parisian households, we found a Dufayel account." (Halbwachs 1908).

13    "The last of the Pailleprés was a charming man whom my father loved very much. He was a deliveryman for Dufayel and perfectly satisfied with his lot. After work, he liked to smoke a pipe while watching Renoir paint in the garden. He said nothing about the painting in progress but happily recounted his day, trying to describe the customers to whom he had delivered Henry II sideboards or one of those imitation wrought iron lanterns with colored glass, so much liked by Courteline." (Renoir [1962] 2020). Note that Georges Courteline integrated a Dufayel collector into his piece *Coco, Coco et Toto*, Paris, Albin Michel, 1905.

14    See, for example, Reynaud (1924, pp. 612–40).

15    "In 1907, [Marcoussis] met Marcelle Humbert in Montmartre. Marcelle had bourgeois aspirations [. . .]. His passion for Marcelle prevailed over his love of painting, so he gave in . . . They could soon leave rue Lamarck for a larger, more bourgeois apartment, 33 rue Delambre, and, the consecration of this gentrification was the purchase of a complete Dufayel furniture set." Lafranchis (1961, p. 52). About Eva Gouel, see Madeline (2023, pp. 64–89).

16    "[. . .]. moved a notch up the social ladder. Mr. Dufayel helped with the acquisition of furniture and a sewing machine. And little by little Florise changed, evolved . . ." Brisson (1902).

17    "Apparently the great Turk uses the services of Dufayel! Good Lord! He must buy women . . . on installments"—and notes that the self-made man has left behind him, "[. . .] a host of charitable works, mutual aid societies [which] receive important bequests. Even in death, Dufayel thought of the humble, the modest, and their difficult future." Anonymous, "Un héros de la Belle Époque. Dufayel ou l'étonnante histoire d'un petit palefrenier devenu, en vingt ans, banquier de la nouveauté et empereur des domes", *La Presse*, 27 January 1948, p. 1 et 3, p. 3.

18    See Daurel (1907, p. 571).

19    See *Gazette des Beaux-Arts*, 1 July 1909, p. 472.

20    See *Les hommes du jour*, 20 December 1913, p. 5, https://gallica.bnf.fr/ark:/12148/cb32787229g/date19131220, accessed on 17 February 2024.

21    "Pleasant attractions: cinema—from 1896—and concert, palmarium, reading room, five o'clock tea" *L'Excelsior*, 1 July 1913, p. 4.

22    "[. . .]. are performed for the most part by the best artists from the *Comédie-Française*, the *Odéon* and the main theaters of Paris . . ." *Comœdia*, 20 July 1909, p. 3.

23    "Free cinema was another of Dufayel's bold innovations" Renoir (Renoir [1974] 2005, pp. 12–14).

24    "Everyone knew that good manners, decidedly abandoned by our old French nobility, had taken refuge in the establishment of [Dufayel], who knew how to make his stores the last salons where people could chat." *La critique indépendante*, 5 April 1908, p. 2.

25    See *Le Figaro*, 24 December 1907, p. 1.

26    "style that synthesizes the times in which we live". See *Gil Blas*, 10 November 1908, np et Gil Blas, 19 November 1902, np.

27    "[. . .]. copy ribbons at Dufayel . . .» Dorgelès (1946).

28    "Painting is beautiful but it's sad, /It's a little lacking in essentials, /Don't count on an artist/To get furnished at Dufayel . . ." "*De place en place» ou Ballade des places de Paris ou Les places de Paris*," music by Adolf Stanislas.

29    "the inconvenience of buying a bed and a piano on the instalment plan." Stein (Stein [1934] 1990, p. 32).

30    See *La Semaine politique et littéraire*, 22 September 1912, p. 8 et *Gino Severini, Tutta la vita di un pittore*. Roma—Parigi, Garzanti, Milan, 1946. Traduction française *La vie d'un peintre*, Paris, Hazan, 2011, p. 33 et 34 (Severini 2011).

31    *Comœdia*, 27 octobre 1908, p. 2; Manneville (2013); *Comœdia*, 25 December 1908, p. 2.

32    "[. . .]. Paris, industrial Paris, business Paris [. . .] contains a good number of painted houses: the movers are in yellow and the color merchants pepper their gigantic signs with the most disparate shades." See Intérim, «Gazette des tribunaux», *Le Figaro*, 12 January 1896, p. 3.

33    "[. . .]. the haunting beauty of commercial inscriptions, posters [. . .]" Text of 1918 cité dans Aragon (1981, p.5).

34    Picasso, *Mandoliniste*, Paris, 1911, Riehen, Fondation Beyeler, Z.IIa.270; La table de l'architecte, Paris, printemps 1912, New York, MoMA, Z.II.1.

35    "[. . .]. especially painted the Henry II sideboard that we had." Cabanne (1974). Par exemple: *Femme au buffet*, Juan-les-Pins, 9 April 1936, Paris, musée national Picasso, MP151.

[36] "[...]. the objects that enter my painting are not like that at all: they are ordinary: a jug, a glass of beer, a pipe, a packet of tobacco. I won't go looking for a rare item that no one has ever heard of . . ." Gilot and Lake (Gilot and Lake [1964] 2006, p. 71).

[37] "furniture, clothing, phonographs, pianos, cookware and works of art." *L'Intransigeant*, 1 March, 1907, p. 4.

[38] "[...]. huge choice of items for gifts such as trinkets, gold items, watches, jewelry, bronze pieces, works of art, small furniture, leather goods, hunting weapons . . ." *L'Aurore*, 25 August 1908, p. 3.

[39] *Bouteille, verre et fourchette*, Paris, hiver 1911–1912, Cleveland, The Cleveland Museum of Art, Z.IIa.320; L'Étagère, Paris, hiver 1911–1912, collection particulière, Z.IIa.310.

[40] "[...]. fake wood, fake marble, glued paper, all these 'ready-made elements'. . ." Daniel-Henry Kahnweiler, *Confessions esthétiques*, Paris, Gallimard, 1963, p. 150.

[41] *Violon: Jolie Eva*, Paris, 1912, Stuttgart, Staatsgalerie, Z.IIa.342.

[42] *Paysage aux affiches*, Sorgues, été 1912, Osaka, National Museum of Art, Z.IIa.353.

[43] "[...]. sign letters and other inscriptions because, in a modern city, the inscription, the sign, the advertising play a very important artistic role . . ." Apollinaire (1913).

[44] "Un mécène montmartrois", *La Critique indépendante*, 23 April 1908, p. 2.

[45] Picasso, *Femme à la mandoline*, Paris, printemps 1910, Suisse, collection particulière, ZII.a.228; Braque, *Femme tenant une mandoline*, Paris, printemps 1910, Munich, Bayerische Staatsgemäldesammlungen, R.71.

[46] Jean-Baptiste-Camille Corot, *Femme portant une toque*, 1950–1955, collection particulière.

[47] See Hélène Seckel-Klein, Picasso collectionneur, Paris, RMN, 1998.

[48] "[...]. the edge of the big city, between cultivated countryside and wasteland, [...] a place favorable to the persistence of ancient peasants and rural traditions . . ." Le Thomas (2016, p. 34).

[49] "You have to know how to be vulgar" Parmelin (1964, pp. 138–39).

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
