# Peer review of "“[…] Un Tout Petit Peu de Dufayel”—Picasso, 1910–1914"

_arts, 1910_

Round 1
Reviewer 1 Report
Comments and Suggestions for Authors
This is a thoughtful, well-researched essay that brings some new material and reflection to what is a a very well-known field of study. I found the argument developed in sections 13-5 stimulating and original.
There are sections - e.g. lines 38-, 43-44, 131-132- that are somewhat unclear. There is also over use of the present tense, in English.
Lines 297, 305 - interesting points re these works and it is a pity they are not illustrated.
Comments on the Quality of English Language
indicated above
Author Response
Thank you for your positive comments. I have thoroughly revised the English version of the article to address the issues you mentioned and to clarify all points. The illustrations that I included address the main issues raised in the article.Reviewer 2 Report
Comments and Suggestions for Authors
Although I read French, the extensive quotes must also be translated to English following the French for those many readers of the journal who do not read French in order to follow the applicability of same. The flow & understanding of the article will be much enhanced by use of both languages.
While the concept & content of this article are very interesting & bear publishing, the extensive problems with physical presentation & organization of same undercut the article enormously. The title neither adequately reflects the content nor invites further reading. The numbered sections (which seem to imply slides? which do not accompany the text) are distracting & inappropriate; delete them. Much sentence word order is not that of English, especially with adverbial phrases, which makes them read very clumsily at best. Some vocabulary choices in English are not the best choices for what the writer intends. Numerous sentence fragments detract from the flow of the information. Paragraphing is semi-inexplicable in terms of related content. Verb tenses are mixed, leading to misunderstandings of timeline & content. Punctuation, especially commas & colons are totally incorrect in English, as is the ordering of usage in quotations.
Comments on the Quality of English LanguageWhile both the language & the usage in English are minimally acceptable to a reader who knows both French & English--one who really wants to follow the content of the article, so is willing to read on in spite of same--frankly, the English is distracting, if not confusing, at best, especially as it would be for any monolingual English-speaker reader of the article, particularly an academic who would be a most likely reader of this journal. Sadly, both so get in the way of the content as to render the article not as interesting as it promises from topic.
Author Response
Thank you for your positive comments. All intext quotes now appear in English. All endnote numbers now appear after punctuation. The article has now been divided into subsections, each headed by its own subtitle.Round 2
Reviewer 2 Report
Comments and Suggestions for Authors
When I suggested in my original review that translation of the quotes to English was indicated for the greater readership, I meant ALONG WITH AND AFTER THE ORIGINAL FRENCH IN THE TEXT . . . but because the content is so interesting and original and this article merits publication, I hereby approve this version without reservation. The other issues are completely resolved.
Author Response
Thanks.